# Impact of the Type of Dialysis on Time to Transplantation: Is It Just a Matter of Immunity?

**DOI:** 10.3390/jcm11041054

**Published:** 2022-02-17

**Authors:** Matteo Righini, Irene Capelli, Marco Busutti, Concettina Raimondi, Giorgia Comai, Gabriele Donati, Maria Laura Cappuccilli, Matteo Ravaioli, Pasquale Chieco, Gaetano La Manna

**Affiliations:** 1Nephrology, Dialysis and Transplantation Unit, IRCCS—Azienda Ospedaliero Universitaria di Bologna, Alma Mater Studiorum, University of Bologna, 40100 Bologna, Italy; matteo.righini5@unibo.it (M.R.); irene.capelli@gmail.com (I.C.); marco.busutti@live.it (M.B.); concettina.raimondi@aosp.bo.it (C.R.); giorgia.comai@aosp.bo.it (G.C.); gabriele.donati@aosp.bo.it (G.D.); 2Nephrology and Dialysis Unit, Santa Maria delle Croci Hospital, 48100 Ravenna, Italy; 3Department of Experimental, Diagnostic and Specialty Medicine (DIMES), University of Bologna, 40126 Bologna, Italy; maria.cappuccilli@unibo.it (M.L.C.); pasquale.chieco@unibo.it (P.C.); 4Department of General Surgery and Transplantation, Policlinico di Sant’ Orsola, 40138 Bologna, Italy; matteo.ravaioli@aosp.bo.it

**Keywords:** peritoneal dialysis, hemodialysis, kidney transplantation, autoimmunity

## Abstract

Background: Renal transplantation represents the therapeutic gold standard in patients with end stage renal disease (ESRD). Still the role of pre-transplant dialysis in affecting time to transplantation has yet to be determined. We wanted to verify whether the type of renal replacement therapy (hemodialysis vs. peritoneal dialysis) affects time to transplantation and to identify clinical features related to the longer time to transplantation. Methods: We performed a retrospective single-center observational study on patients who had received a transplant in the Bologna Transplant Unit from 1991 to 2019, described through the analysis of digital transplant list documents for sex, age, body mass index (BMI), blood group, comorbidities, underlying disease, serology, type of dialysis, time to transplantation, Panel Reactive Antibodies (PRA) max, number of preformed anti Human Leukocyte Antigens (HLA) antibodies. A *p*-value < 0.05 was considered statistically significant. Results: In the 1619 patients analyzed, we observed a significant difference in time to transplant, PRA max and Preformed Antibodies Number between patients who received Hemodialysis (HD) and Peritoneal dialysis (PD). Then we performed a multiple regression analysis with all the considered factors in order to identify features that support these differences. The clinical variables that independently and directly correlate with longer time to transplantation are PRA max (*p* < 0.0001), Antibodies number (*p* < 0.0001) and HD (*p* < 0.0001); though AB blood group (*p* < 0.0001), age (*p* < 0.003) and PD (*p* < 0.0001) inversely correlate with time to transplantation. Conclusions: In our work, PD population received renal transplants in a shorter period of time compared to HD and turned out to be less immunized. Considering immunization, the type of dialysis impacts both on PRA max and on anti HLA antibodies.

## 1. Introduction

Hemodialysis (HD) and Peritoneal Dialysis (PD) are the two most common forms of renal replacement therapy, life-saving treatment for patients with End Stage Renal Disease (ESRD). Despite there being contraindications for each treatment, nowadays the choice of the kind of treatment depends on several features, mainly the specific experience of the Clinical Unit and the patient’s choice [1]. Although PD is a well-established treatment modality it is underused in Western countries, with a prevalence in Italy of 15% [2,3]. Recently, several studies showed a relative survival advantage for patients receiving PD lasting one to two years after dialysis initiation [4,5,6,7,8]. Renal transplantation represents the therapeutic “gold standard” in patients with ESRD ensuring better outcomes compared with dialysis both in patient survival and quality of life [2,9,10,11]. The role of pre-transplant dialysis choice in affecting transplant outcomes has been the subject of long-standing interest [12,13,14,15]. For those who could not stand the prospect of living donor transplantation, being on the waiting list for kidney transplantation from a deceased donor is a vital choice. According to CNT (National Transplant Centre) data, in our country a patient who signs up for renal transplant waiting list normally waits 3.3 years before getting a transplant [16]. There are several known demographic and clinical factors that affect time to transplantation (age, blood group, and level of antibody sensitization) [14] but many other factors are implied.

We wanted to verify whether the type of renal replacement therapy (HD vs. PD) affects time to transplantation and we wanted to identify clinical features related to longer time to transplantation.

## 2. Materials and Methods

We performed a retrospective study on patients who have received a transplant in the Bologna Transplant Unit from 1991 to 2019, described through the analysis of digital transplant list documents for sex, age, BMI, blood group, hypertension, diabetes, cardiovascular disease, neoplastic disease, underlying disease, serology (HBV, HCV, HIV, CMV, Toxo, EBV, LUE), type of dialysis, time to transplantation, Panel Reactive Antibodies (PRA max), and number of preformed anti Human Leukocyte Antigens (HLA) antibodies. Immunologic data were collected regarding typing of HLA, PRA max that represent the maximal value of PRA in the considered waiting time to transplant. PRA was expressed as the percentage of lymphocyte panel members against which the patient’s serum reacts and thus against which the patient has HLA class I or II antibodies. Since 2012 all the PRA was tested with the complement-dependent cytotoxicity test (PRA-CDC) for monitoring the degree of immunization in kidney transplant candidates on active waiting lists, after that year patients were tested with the use of Labscreen PRA class I and II on a Luminex platform. The number of preformed antibodies expressed the quantity of specific preformed HLA class I or II that the recipient patient presented. Levels of normalized, mean fluorescence intensity >1000 were considered to be positive.

We excluded patients who had received a previous transplant, who were recorded in the national hyperimmune program supposing that those patients were too immunized and they have been on the waiting list a very long time, patients transplanted or signed up in pre-emptive modality, and patients who received a combined transplant (Figure 1).

We used survival curves to analyze time to transplantation with transplantation as an end point.

Data were taken from the digital medical records and was imported to create a database specifically for the study. The study was approved by the Local Ethical Committee.

## 3. Statistical Analysis

The aim of statistical analysis in this observational study was to find factors associated with time to transplantation. The proper test of statistical significance depends on the nature of the examined variables. Student’s *t*-test or ANOVA followed by Tukey post-hoc test, corrected for heteroscedasticity, when necessary, were used for real outcome variables. For non-parametric outcomes we used Mann-Whitney test or Kruskal-Wallis ANOVA followed by Dunn’s test for pair-wise comparisons. Categorical data were analyzed using contingency tables and χ^2^. Linear associations using one or more covariate were explored with linear regression. Survival curves were computed by Kaplan-Meier estimate [17] and compared by a log-rank test. Continuous variables are presented as mean ± Standard Deviation or as median and Interquartile interval (IQR) when appropriated. All statistical tests were two-tailed, and we used JMP 13 (SAS Institute Inc., Cary NC, USA) for data management and analysis. *p*-values were not corrected for multiplicity and the findings should be interpreted as exploratory. A *p*-value < 0.05 was considered statistically significant.

## 4. Results

We considered in our analysis 2343 patients who received a transplant in Bologna from 1991 to 2019. According to exclusion criteria (Figure 1), we analyzed 1619 patients. As is shown in Table 1, our population presented 1053 (65%) men, mean age at transplantation was 61.9 ± 12.4 years, mean years of dialysis before transplantation were 4.4 ± 6.5 years (1618.34 ± 2376.75 days), 554 (34.2%) patients had hypertension, 214 (13.2%) had diabetes, 79 (4.9%) had cardiovascular disease, 18 (1.1%) had a history of neoplastic disease. Mean BMI was 24.1 ± 3.5. There were 1347 patients who received HD and 271 (16.7%) received PD, latency from the beginning of dialysis to the enrolment on the transplant waiting list resulted in 354.5 days (IQR 108.6 days); 901 (55.6%) patients presented with HbsAb; 1037 (64%) patients had positive CMV IgG; 653 (40.3%) patients had Blood Group A; 180 (11.1%) had B; 705 (43.5%) had 0; and only 78 (4.8%) had AB. In our cohort 169 (10.4%) patients had IgA nephropathy; 161 (9.9%) had hypertensive nephropathy; 328 (20.3%) presented with Polycystic Kidney Disease; 153 (9.4%) had Diabetic Nephropathy; 374 (23.1%) patients presented with other nephropathies while 433 (26.7%) patients had an unknown diagnosis. In total, 554 patients presented with hypertension. Mean time to transplantation was 2.4 ± 2.6 years (889.3 ± 945.7 days), PRA max resulted in 21.04 ± 31.5 and number of preformed antibodies resulted in 4.6 ± 12.9.

As previously described, we aimed to analyze whether the type of dialysis could influence time to transplant. Indeed, we observed a significant difference in time to transplant (933 ± 25.6 days in HD and 667.3 ± 57.1 days in PD, *p* < 0.001), PRA max (23.1 ± 0.9 in HD and 11.4 ± 1.9 in PD, *p* < 0.001) and Preformed Antibodies Number (5.1 ± 0.3 in HD and 2 ± 0.7 in PD, *p* < 0.001) between patients who received HD and PD. Then we performed a multiple regression analysis with all the considered factors in order to identify features that support these differences. In Table 2 the clinical variables that independently and directly correlates with longer time to transplantation are listed: PRA max (*p* < 0.0001), Antibodies number (*p* < 0.0001) and HD (*p* < 0.0001). Other features inversely correlate with time to transplantation: AB blood group (*p* < 0.0001), age (*p* < 0.003) and PD (*p* < 0.0001).

Assuming that immunization is a key factor in determining time to transplant, though we analyzed our PRA max data according to the model that Bostock et al. used [18] (Figure 2). Then we performed a multiple regression on PRA max: the only parameters that correlate with PRA max values are the type of dialysis (*p* = 0.0015), AB blood group (*p* = 0.0017) and BMI (*p* = 0.0079) (Table 3). Another way to describe immunization is through antibodies’ expression: as shown in Table 4, in performing a multiple regression analysis on antibodies number we evidenced that the only features that correlate with a higher number of pre-formed antibodies were age (*p* = 0.0001) and type of dialysis (*p* = 0.0004).

Considering time to transplantation to be our first goal we represent our results in Figure 3. Then we created subgroups according to Age, BMI, Blood Group, and Diagnosis, exploring whether these differences persisted in all the subgroups. Data are presented in Table 3.

### 4.1. Age

Considering age, we analyzed the population divided into 5 groups: Group 1 Age ≤ 45; Group 2 45 < Age ≤ 55; Group 3 55 < Age ≤ 65; Group 4 65 < Age ≤ 75; Group 5 Age > 75. In the HD population we found significant differences between group 3 (1117 ± 49.4 days) and 5 (680.4 ± 72.9 days, *p* < 0.0001), and between group 3 and 4 (854.5 ± 50.4 days, *p* = 0.002).

In the PD population we pointed out significant differences only between group 2 (815.1 ± 84.4 days) and 4 (521.1 ± 60.9 days, *p* = 0.04).

Considering age to be a factor that independently correlates with Antibodies number, we performed the same analysis, showing that in the HD population there existed a significant difference between group 2 (7.2 ± 0.9) and group 5 (1.9 ± 1, *p* < 0.001); group 1 (6.8 ± 1.2) and group 5 (*p* < 0.01); group 3 (5.9 ± 0.7) and group 5 (*p* < 0.01); and group 2 and group 4 (3.8 ± 0.7, *p* < 0.02). Conversely, in the PD population we evidenced no differences.

### 4.2. BMI

Considering BMI, we analyzed the two populations divided into 4 groups: Group 1 BMI ≤ 20, Group 2 20 < BMI ≤ 25, Group 3 25 < BMI ≤ 30, Group 4 BMI > 30. We did not found any significant difference among groups both in HD as in PD.

Since BMI resulted in an independent factor that influenced PRA max, we performed the same analysis showing that in the HD group there existed significant differences between group 1 (26.1 ± 33.5) and group 2 (20.2 ± 30.6, *p* < 0.048), and between group 1 and group 3 (19.2 ± 28.7, *p* < 0.02). In the PD population we highlighted no differences among groups.

### 4.3. Blood Group

Considering blood group, we analyzed the population according to the 4 groups: A, AB, B, O. In the HD population we evidenced significant differences between 0 (1124.5 ± 39.7 days) and AB (426 ± 131.7 days, *p* < 0.0001), B (957.3 ± 79.5 days) and AB (*p* = 0.003), 0 and A (765.2 ± 42.3 days, *p* < 0.0001). The difference between A and AB is nearly significant (*p* = 0.06). In the PD group differences occurred between O (840.3 ± 59 days) and AB (433.9 ± 122.9 days, *p* = 0.01) and between O and A (590.1 ± 54 days, *p* = 0.01).

Since blood group represented a category that influence the PRA max, we performed the same group analysis but we did not evidence differences both in HD as in PD.

Between HD and PD there existed in our whole population a significant difference in PRA max for the A group (mean HD PRAmax 20.7 ± 1.4, mean PD PRAmax 8.9 ± 2.9, *p* = 0.0004), AB group (mean HD PRAmax 22.2 ± 4.2, mean PD PRAmax 3 ± 6, *p* = 0.01) and O group (mean HD PRAmax 25.9 ± 1.4, mean PD PRAmax 15.1 ± 3.4, *p* = 0.003).

### 4.4. Diagnosis

Considering diagnosis, we analyzed the population divided into 6 groups: IgA nephropathy, Hypertensive, ADPKD, Diabetic, Other nephropathies, Unknown origin.

Both in the HD as in the PD population we evidenced no differences among groups.

Considering that ESA could have substantially reduced the use of blood transfusions, we divided our population in two groups, before 2000 (old group) and after 2000 (new group), considering that ESA were introduced in 1990 and in the following years achieved global distribution. We compared the Antibodies numbers of two groups showing that there was no significant difference (HD group *p* = 0.068, PD group *p* = 0.290) but the statistical difference remained if we compared the HD old group vs. PD old group (*p* = 0.029) and the HD new group vs. PD new group (*p* = 0.018).

## 5. Discussion

Prolonged dialysis exposure casts a long shadow and negatively impacts patient and graft survival even after transplantation [19,20,21]. Despite similar results in terms of outcome, PD is far less used as renal replacement therapy compared to HD [2,3]. Decisions regarding dialysis choice should be individualized, considering several important outcomes including patient survival and quality of life. A study conducted by Heaf et al. analyzed the relative survival of PD compared to HD, showing that PD has a relatively better prognosis for younger and non-diabetic patients, and for no subgroup was worse than HD; one possible cause is the better preservation of residual renal function in PD [22,23]. A shorter time to transplant could be one of the drives that lead the patient’s choice. Several studies analyzed kidney transplant outcomes according to dialysis modality; most studies revealed that PD was associated with shorter time on dialysis, better graft and patient survival [24,25,26,27,28,29,30,31].

Recent works reported that the transplant list waiting time is reduced in PD compared to those on HD [13,15] and the results in our cohort confirm this data. What causes these differences is yet to be proven: some explanations can be hypothesized.

As reported by several studies, sensitization is a known barrier to transplantation and sensitized patients have substantially longer waiting times; the breadth of sensitization against HLAs is routinely monitored in wait-listed patients with ESRD using panel reactive antibody (PRA) assays [32,33,34,35]. Previous transplant, in addition to pregnancies and blood transfusions, is a known cause of immune sensitization against “non-self” human leukocyte antigens (HLAs). Moreover, PRA is an independent predictor of mortality in wait-listed kidney transplant candidates [36]. There is good epidemiologic evidence showing a direct relationship between pretransplant PRA levels and adverse graft outcomes [37,38,39]. Our study confirms that an elevated PRA is associated with longer time to transplant, but also shows that even patients on PD have a lower mean PRA compared to those on HD (*p* < 0.0001).

Considering anti-HLA antibodies, our study shows that they are related to an increased time to transplantation, as supported by a large cohort study, where the authors suggest that anti-HLA antibodies are associated with an increased sensitization and mortality in wait-listed kidney transplant candidates [36]. In our cohort, PD patients presented less anti-HLA antibodies than HD patients (*p* < 0.001). A reason may lie in the fact that HD patients had a greater tendency to anemia and more likely may require blood transfusions, which can determine an increase in panel reactive antibody percentage [15,40]. Other explanations may be further investigated. Our results suggest that BMI is a factor that influences kidney transplant recipient immunization. Lots of studies have investigated obesity in kidney transplant recipients, recognizing its importance as a risk factor for chronic allograft dysfunction, exposing to a major risk of delayed graft function and considering some cases an exclusion criteria for transplantation [41,42,43]. Several mechanisms exert negative metabolic effects of raised BMI and adiposity, and recent studies demonstrated that higher BMI was associated with higher inflammation, that correlates with mortality [44,45,46].

Indeed, in our cohort, PD patients presented a significantly reduced waiting time for kidney transplant even eliminating the impact of immunization: so we have to wonder that other non-immunological factors may have an impact on this result.

Patients who undergo PD tend to be more empowered, to have a strong social support network and to pursue their care plan by themselves [47]. Moreover, our PD group had a lower pretransplant dialysis vintage compared with the HD group (PD 225–271.8 vs. HD 383–118.3, *p* = 0.06), leading to a different exposure to dialysis-related immune dysfunction. Nonetheless, the latency from the beginning of dialysis to the enrolment on the transplant waiting list resulted in differences between the PD group and the HD group, though nearly significant (*p* = 0.06). This could once more be explained by the strong social support network of patients who chose PD.

Age is a considering factor in transplant recipients and up to twenty years ago, lots of patients would have been excluded due to being elderly. The “old to old” allocation system in the Eurotransplant [48,49] community has shown to be effective in increasing the number of transplants; thus, in our population 45 to 65 years old patients were the most represented age groups, therefore they present a longer mean time to transplantation. These results are in line with literature [50].

Regarding the relationship between blood groups and time to transplant, Chang et al. [51] reported that blood group AB have the highest likelihood of deceased donor transplantation, followed by patients with blood groups A, O and B; within blood groups, the likelihood of transplantation was inversely related to the level of sensitization (PRA max). Our study confirms previous work showing longer waiting times for blood group O patients [14,52] and that AB recipients were more likely to receive deceased donor kidneys [53]. Thus, AB patients represent a small population, though not sufficient to draw conclusions on time to transplant.

Our study presents some limitations: in addition to the retrospective nature of the analysis due to the long-considered time and to patients’ dialysis center variability, the precise number of blood transfusions was not available. Since the study covers 30 years of experience, variability in dialysis practice (HD vs. PD) between different referral centers made the population less homogeneous.

## 6. Conclusions

In our work, the PD population received renal transplant in a shorter period of time compared to the HD, turned out to be less immunized, considering immunization to be the number of antibodies and PRA. Time to transplant is mostly a matter of immunity but other factors can influence it, such as age, blood group and type of dialysis. Considering immunization, the type of dialysis impacted both on PRA max (together with BMI and blood group) as on anti HLA antibodies (together with age). Our study supports the choice of PD for patients who can afford it, particularly for those who would like to receive a kidney transplant.

## Figures and Tables

**Figure 1 jcm-11-01054-f001:**
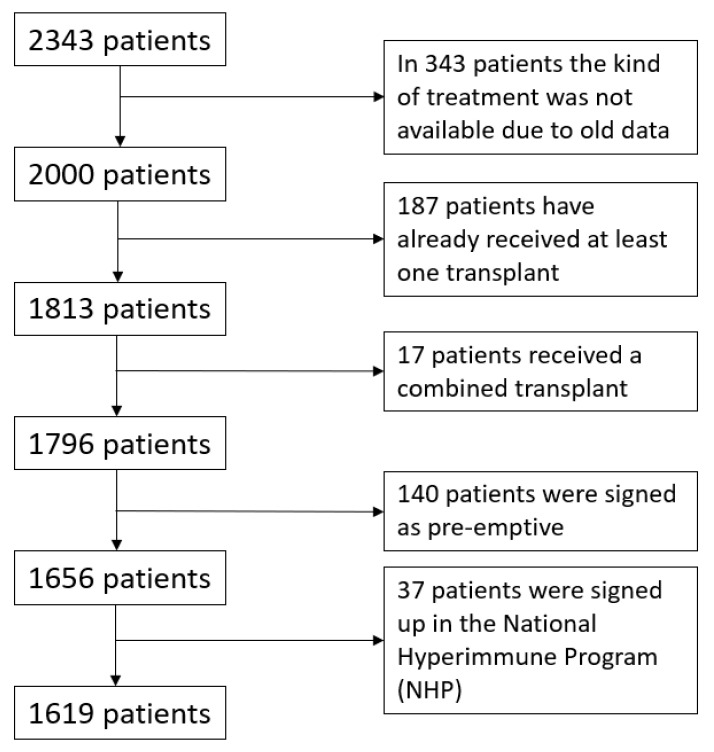
Patients who received a kidney transplant in Bologna Transplant Unit between 1991 and 2019 and exclusion criteria.

**Figure 2 jcm-11-01054-f002:**
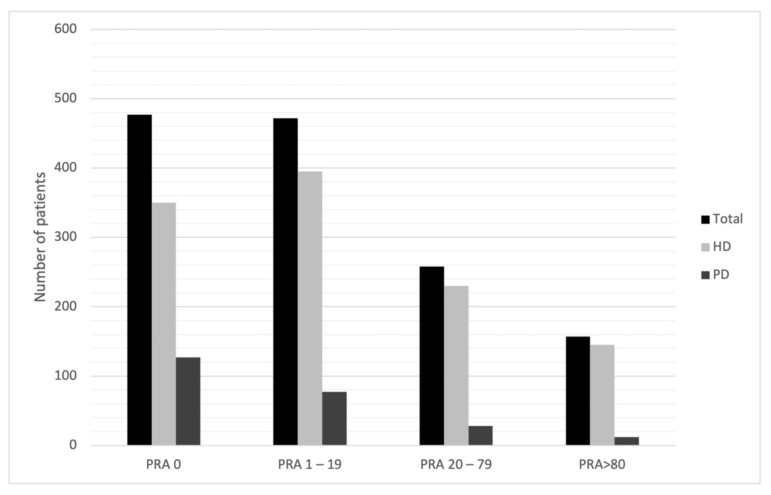
Patients divided according to PRA max. The classes were selected according to the study conducted by Bostock IC et al. [18]. HD: hemodialysis. PD: peritoneal dialysis.

**Figure 3 jcm-11-01054-f003:**
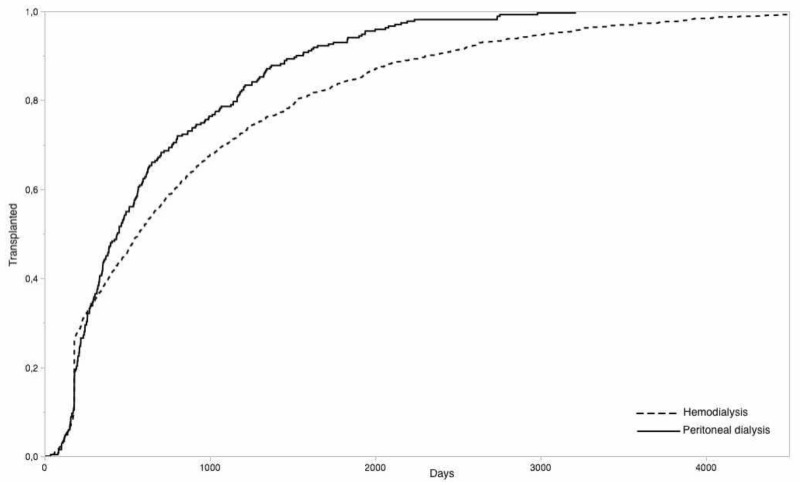
Patients who received kidney transplant divided by type of dialysis. Those who received PD (271 patients) reached the goal (kidney transplant) far earlier than those who received HD (1347 patients) (Log-Rank ChiSquare 21.99, *p* < 0.0001, Wilcoxon ChiSquare 5.77, *p* = 0.0163).

**Table 1 jcm-11-01054-t001:** Features of the study population and according to the type of dialysis.

	Population	Hemodialysis	Peritoneal Dialysis	*p* Value
No. of patients	1619	1347 (83.2%)	271 (16.8%)	
Sex Male Female	1053 (65%) 565 (35%)	898 (66.7%) 449 (33.3%)	155 (57.2%) 116 (42.8%)	0.002 *
Age (years)	61.9 ± 12.4	61.9 ± 12.3	62.2 ± 12.5	ns
Blood group A B AB O	651 (40.2%) 179 (11.1%) 76 (4.7%) 705 (43.5%)	533 (39.6%) 150 (11.1%) 54 (4%) 605 (44.9%)	118 (43.5%) 29 (10.7%) 22 (8.1%) 100 (36.9%)	0.007 *
BMI	24.1 ± 3.5	23.9 ± 3.5	24.7 ± 3.4	0.001 *
Diabetes	214 (13.2%)	182 (13.5%)	32 (11.8%)	ns
Hypertension	554 (34.2%)	452 (33.6%)	102 (37.6%)	ns
Cardiovascular disease	79 (4.9%)	56 (4.2%)	23 (8.5%)	ns
Neoplastic disease	18 (1.1%)	15 (1.1%)	3 (1.1%)	ns
Nephropathy Unknown IgA nephropathy Hypertensive ADPKD Diabetic Other	433 (26.7%) 169 (10.4%) 161 (9.9%) 328 (20.3%) 153 (9.4%) 374 (23.1%)	366 (27.2%) 138 (10.2%) 127 (9.4%) 284 (21.1%) 126 (9.3%) 306 (22.7%)	67 (24.7%) 31 (11.4%) 34 (12.5%) 44 (16.2%) 27 (10%) 68 (25.1%)	ns
Prior time of dialysis until listing for transplantation Med [IQR]	354.5 [108.6] days	383 [118.3] days	225 [271.8] days	ns
HCV IgG	61 (3.8%)	58 (4.3%)	3 (1.1%)	ns
HBsAb	901 (55.6%)	753 (55.9%)	148 (54.6%)	ns
HbcAb	220 (13.6%)	186 (13.8%)	34 (12.5%)	ns
CMV IgG	1037 (64.1%)	839 (62.3%)	198 (73.1%)	ns
PRA max (%) 0 1–19 20–79 >80	477 (29.5%) 472 (29.1%) 258 (15.9%) 157 (9.7%)	350 (26%) 395 (29.3%)230 (17.1%) 145 (10.8%)	127 (46.9%) 77 (28.4%) 28 (10.3%) 12 (4.4%)	

Features of the study population described as Sex, Age at the time of transplantation, BMI, underlying nephropathy, prior time of dialysis until listing for transplantation, serology for HCV, HBV and CMV. Cardiovascular disease was considered as patients who had heart failure or previous myocardial infarction. All the variables were compared through Student’s *t*-test or ANOVA followed by Tukey post-hoc test, corrected for heteroscedasticity. BMI: Body Mass Index. ADPKD: Autosomic Dominant Polycystic Kidney Disease. * A *p* value was considered significant when *p* < 0.05.

**Table 2 jcm-11-01054-t002:** Multiple regression for Time to transplant.

Source	LogWorth		*p* Value
PRAmax	17.382	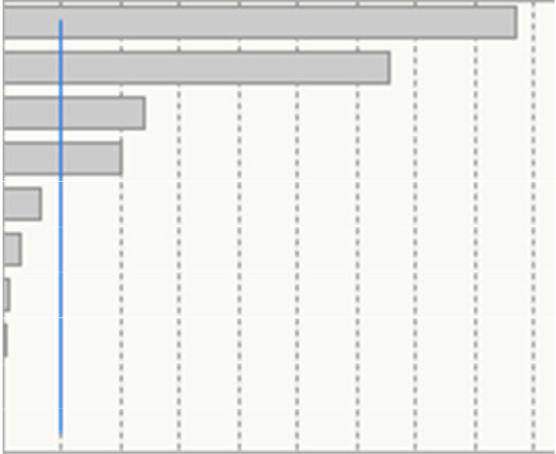	0.00000
Blood group	13.148		0.00000
Type of dialysis	4.808		0.00002
Antibodies Number	4.080		0.00008
Age	1.332		0.04654
CMV IgG	0.611		0.24496
Diagnosis	0.200		0.63037
BMI	0.187		0.65032
Sex	0.037		0.91757
HBsAb	0.023		0.94883

We can observe that the variables that directly correlate with time to transplantation are PRA max, Antibodies Number and HD; factors that inversely correlate are AB blood group, age and PD. These results are confirmed even in the Multivariate analysis. PRA = Panel Reactive Antibodies. HD = Hemodialysis. PD = Peritoneal Dialysis.

**Table 3 jcm-11-01054-t003:** Multiple regression for PRA max.

Source	LogWorth		*p* Value
Type of dialysis	2.831	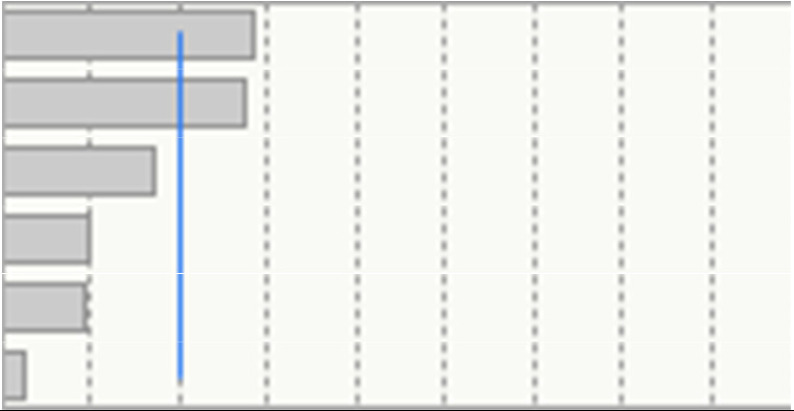	0.00148
Blood group	2.759		0.00174
BMI	1.733		0.01848
Age	0.989		0.10251
HBsAb	0.959		0.11001
CMV IgG	0.249		0.56355

Factors that correlate with elevated PRA max are Type of dialysis, AB blood group and, slightly, BMI. These factors are confirmed even in the multivariate analysis.

**Table 4 jcm-11-01054-t004:** Multiple regression for Antibodies Number.

Source	LogWorth		*p* Value
Age	2.771	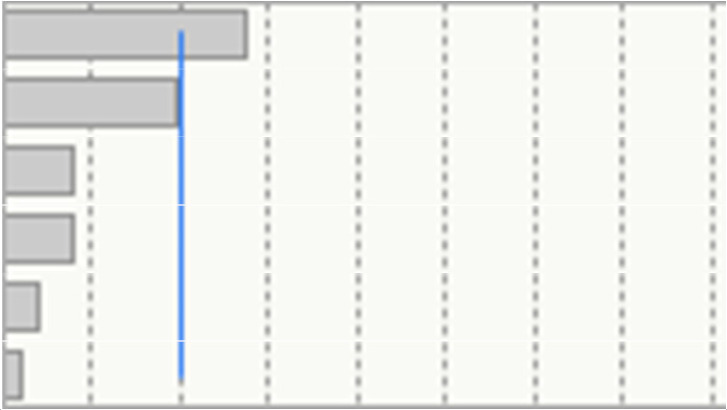	0.00169
Type of dialysis	1.985		0.01034
BMI	0.812		0.15432
HBsAb	0.784		0.16447
Blood Group	0.407		0.39161
CMV IgG	0.197		0.63481

Factors that correlate with Antibodies Number are Type of dialysis and Age. These factors are confirmed even in the multivariate analysis.

## Data Availability

Data can be found on the Donor Manager transplant program.

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
