# Peer review of "Impact of the Type of Dialysis on Time to Transplantation: Is It Just a Matter of Immunity?"

_jcm, 2022, doi:10.3390/jcm11041054_

Round 1
Reviewer 1 Report
Interesting paper.
Author Response
Dear reviewer,
thank you very much for your revision.
We have revised our English language as suggested.
Greetings
Matteo Righini
Reviewer 2 Report
The manuscript is clearly written. It was shown that PD patients received renal transplant in a shorter period of time compared to HD and were less immunized.
The key message is clear. I have some minor remarks.
- The percentage of comorbidities is relatively low. Please explain it.
- Please exchange comas with dots in non-integers.
- The sentences from line 136-137 could be moved to first part of Materials and Methods.
- First sentence of Abstract should be rephrased.
Author Response
Dear reviewer,
thank you very much for your review, we think it could have really improved our work.
- The percentage of comorbidities is relatively low: data were recorded in a long period of time and it wasn't always possible to collect the whole patient's history, however we review all of our patients history and considered as diabetic those who took medication for diabetes, hypertension those who assumed anti hypertensive agent and cardiovascular disease in those who had heart failure or had myocardial infarct.
- We exchange all the comas with dots when indicated
- is this the sentence that you intend? "Other features inversely correlate with time to transplantation: AB blood group (p < 0.0001), age (p <0.003) and PD (p < 0.0001)". We think that this is a result though we describe it in this part of the paper.
- We rephrased the first sentence of the abstract as you suggest
Thank you very much for your suggestions
Greetings
Matteo Righini
This manuscript is a resubmission of an earlier submission. The following is a list of the peer review reports and author responses from that submission.
Round 1
Reviewer 1 Report
The manuscript is clearly written. It was shown that PD patients received renal transplant in a shorter period of time compared to HD and were less immunized.
My major remark is that is not shown if comorbidities have any impact on a waiting time. Next question is whether acute dialysis start has any impact on time of dialysis before transplantation.
Minor:
It was stated that "continuous variables are presented as mean ± Standard Deviation or as median and Interquartile interval (IQR)". There is Med [IC] in Table 1.
Prior time of dialysis in Table 1 regards time to transplantation or enrollment in waiting list? It is unclear.
PD and DP are used alternately, please use PD.
Author Response
Dear reviewer,
thank you very much for your comments.
According to your suggestions:
- unfortunately it was difficult to determine whether some comorbidities has an impact on time to transplantation, the cohort was really heterogeneus and we have data only about nephropathies who were not significative.
- due to the heterogeneus cohort of patients, that come from all over Italy and world it wasn't possible to determine whether acute dialysis start has any impact on time of dialysis before transplantation. It would have been a great data, thus we'd like to make studies above it.
- IC was corrected as you suggest to IQR in Table 1.
- Prior time of dialysis represents time that patients spend on dialysis before being enrolled in transplant waiting list.
- as you suggested, we correct DP in PD in the whole document.
Thank you very much, available for every questions
Greetings
Reviewer 2 Report
It is an interesting study, which describes the different accesses of patients on peritoneal dialysis or hemodialysis for kidney transplantation.
However, I have some comments that should be addressed.
- First sentence in Abstract: affecting transplant outcomes. I think you do not investigate transplant outcomes. I would use the term waiting time instead of transplant outcomes
- Abstract: line 23/24. Remove sentence, it is the repeating of the last sentence of the introduction. Overall, the abstract should be formulated more tightly.
- We performed a retrospective single-center observational study
- Statistical Analysis should be mentioned in the abstract: According to ... test, of the 1619 patients analyzed...
- Please write out Abbreviations, if used the first time (also in abstract), e.g. PRA
- page3/10: 4,4 +/-6,5 (1618,34 +/-2376,75). Please specify the content in the brackets, days?).
- Table 1 is not self-explanatory. Which statistical test was used to compare the groups. Please explain in a legend and the tables in more detail.
- Please specify in table 1: Prior time of dialysis "until listing for Transplantation?" It is confusing whether it is not the time to transplantation the first time you read it and can only be concluded from the text.
- Table 2: Please use a legend and explantation.
- Figure 2: Please include Patient at risk (Patient at dialysis) under the figure. How many patients died on the waiting list in the HD and in the DP group. This should be also included in the calculation and mentioned in the text.
Author Response
Dear reviewer,
thank you very much for your comments and suggestions.
- corrected as suggested.
- corrected as suggested, we make the abstract shorter
- corrected as suggested
- corrected as suggested
- corrected as suggested
- corrected as suggested, the content in brackets was referred to days
- thank you for your suggestion. We added a table legend in order to better clarify the informations
- exactly, that paramether is referred to that. We have corrected as requested.
- we introduced a legend for Table 2 and we inserted an explanation of the table in order to better clarify.
- thank you for your comment. Unfortunately this data in unavailable because we considered only patients who have been transplanted and not those who have died on waiting list.
Thank you for the review process and for the suggestions. Available for every question
Greetings